# Determinants of adolescents' Health-Related Quality of Life and psychological distress during the COVID-19 pandemic

Roxane Dumont[1], Viviane Richard[1], Hélène Baysson[1,2], Elsa Lorthe[1], Giovanni Piumatti[3,4], Stephanie Schrempft[1], Ania Wisniak[1,5], Rémy P. Barbe[6], Klara M. Posfay-Barbe[7], Idris Guessous[2,8‡], Silvia Stringhini[1,2,9‡]*, on behalf of the Specchio-COVID19 study group[¶]

1 Unit of Population Epidemiology, Division of Primary Care Medicine, Geneva University Hospitals, Geneva, Switzerland, 2 Department of Health and Community Medicine, Faculty of Medicine, University of Geneva, Geneva, Switzerland, 3 Institute of Public Health, Faculty of BioMedicine, Università della Svizzera Italiana, Lugano, Switzerland, 4 Fondazione Agnelli, Turin, Italy, 5 Institute of Global Health, Faculty of Medicine, University of Geneva, Geneva, Switzerland, 6 Division of Child and Adolescent Psychiatry, Department of Woman, Child, and Adolescent Medicine, Geneva University Hospitals, Geneva, Switzerland, 7 Division of General Pediatrics, Department of Woman, Child, and Adolescent Medicine, Geneva University Hospitals, Geneva, Switzerland, 8 Division and Department of Primary Care Medicine, Geneva University Hospitals, Geneva, Switzerland, 9 University Center for General Medicine and Public Health, University of Lausanne, Lausanne, Switzerland

☯ These authors contributed equally to this work.
‡ IG and SS also contributed equally to this work.
¶ Membership of the Specchio-COVID19 study group is provided in the Acknowledgments.
* silvia.stringhini@hcuge.ch

**Data Availability Statement:** Due to the fact that we are reporting information on a relatively small sample of adolescents from a specific geographic

## Abstract

### Background

We examined the determinants of adolescents' Health-Related Quality of Life (HRQoL) and psychological distress (self-reported and parent-reported) during the COVID-19 pandemic, using a random sample of the population of Geneva, Switzerland.

### Methods

Data was drawn from participants aged 14–17 years, who participated with their families to a serosurvey conducted in November and December 2020. Adolescents' HRQoL was evaluated using the validated adolescent-reported KIDSCREEN-10 and parent-reported KINDL® scales. Psychological distress was assessed with self-reported sadness and loneliness, and using the KINDL® emotional well-being scale. Using generalized estimating equations, we examined the role of socio-demographic, family and behavioural characteristics in influencing adolescents' mental health status and wellbeing.

### Results

Among 240 adolescents, 11% had a low HRQoL, 35% reported sadness and 23% reported loneliness. Based on parents' perception, 12% of the adolescents had a low HRQoL and 16% a low emotional well-being. Being a girl (aOR = 3.20; 95%CI: 1.67–6.16), increased

location, our data contains potentially identifying information, despite all precautions taken. Therefore, the study steering committee members decided to make these data accessible to researchers who meet the criteria for access to confidential data upon reasonable request for data sharing to the Unit of Population Epidemiology (uep@hcuge.ch). All requests will be evaluated by the Data Access Committee and approved on the basis of their scientific quality.

**Funding:** The Specchio-COVID19 study was funded by the Swiss Federal Office of Public Health, the General Directorate of Health of the Department of Safety, Employment and Health of the canton of Geneva, the Private Foundation of the Geneva University Hospitals, the Swiss School of Public Health (Corona Immunitas Research Program) and the Grangettes Foundation. Funding sources are from private foundations whom do not provide grant/reference numbers. The funders had no role in study design, data collection and analysis, decision to publish, or preparation of the manuscript.

**Competing interests:** The authors have declare that they have no competing interests.

**Abbreviations:** COVID-19, Coronavirus Disease 2019; GEE, Generalized estimating equation; HRQoL, Health-Related Quality of Life; OECD, Organisation for Economic Co-operation and development.

time on social media (aOR = 2.07; 95%CI: 1.08–3.97), parents' *average to poor* mood (aOR = 2.62; 95%CI: 1.10–6.23) and *average to poor* household financial situation (aOR = 2.31; IC95%: 1.01–6.10) were associated with an increased risk of sadness. Mismatches between adolescents' and their parents' perception of HRQoL were more likely for girls (aOR = 2.88; **95%CI: 1.54–5.41)** and in households with lower family well-being (aOR **= 0.91; 95%CI: 0.86–0.96**).

## Conclusions

A meaningful proportion of adolescents experienced low well-being during the second wave of COVID-19, and average well-being was lower than pre-pandemic estimates. Adolescents living in underprivileged or distressed families **seemed particularly affected**. Monitoring is necessary to evaluate the long-term effects of the pandemic on adolescents.

## Background

The COVID-19 pandemic and the measures put in place by public health authorities to contain its spread have caused significant disruptions in daily life and raised concerns for mental health of the entire population. A growing body of literature shows that the mental health of adolescents has deteriorated during the pandemic, in particular during lockdowns [1, 2]. Indeed, adolescence is characterized by important psychological and physical changes, and greater vulnerability to external events [3]. During this sensitive stage of life, the importance of peer-interactions increases in parallel with a desire for greater autonomy from parents [4].

Because of increased difficulties to meet these developmental needs, adolescents may have been particularly affected by the pandemic [1]. Educational disruptions, the widespread use of distance learning, the introduction of social distancing, or cancellation of extra-curricular activities combined with the general stay-at-home message, have led to significant changes in daily life, less time spent with peers, and more with the family [5]. These unprecedented circumstances also resulted in changes in health behaviours that could be detrimental to health such as an increase in screen time [6] and "junk food" consumption [7], together with a decrease in physical activity [8, 9]. Adolescents may have been overly burdened by these life changes along with the worrying pandemic environment [10]. Adolescents living in underprivileged families or whose parents present a decreased mental or health state could be particularly vulnerable as they may experience more external stressors, along with less resources to adapt to the changes induced by the pandemic [11, 12].

Throughout the pandemic, COVID-19-related health policies were less stringent in Switzerland compared with the OECD countries [13]. After a 10 week semi-lockdown in spring 2020, schools remained open, while sanitary measures, such as limitations on size of gatherings, and the requirement to wear masks in public spaces were maintained. Inland travel was never limited; non-essential shops and sport facilities intermittently closed according to the local COVID-19 incidence, which was very high over the study period [14]. Assessing adolescents' well-being and psychological distress in this specific context is important to compare the impact of different measures at the international level. A better understanding of adolescents' well-being during these challenging times is also useful given the uncertainties related to the progression of the pandemic.

## Methods

In this study, we aimed to assess the determinants of adolescent's Health-Related Quality of Life (HRQoL) and psychological distress (self-reported and parent-reported) during the second COVID-19 wave, using a random sample of the population of Geneva, Switzerland.

### Survey design

A serological study was conducted during the second COVID-19 wave, between November 23rd and December 23rd 2020, in the general population of the canton of Geneva, Switzerland [15]. A random sample of families with children or adolescents drawn from state registers was invited to participate and do an anti-SARS-CoV-2 serology. Among the 3510 households invited, 597 (17%) families participated. Of these, 194 families had one or two adolescents (siblings) aged 14 to 17, who were included in the current study. This age range was chosen for adolescents to be mature enough to autonomously answer the study questionnaire. All participants signed a written informed consent and the study was approved by the regional ethics committee (ID: 2020–0088).

Each family designated a "referent parent" who completed a comprehensive health and socio-demographic questionnaire about themselves and about each of their children. Additionally, adolescents were asked to complete a paper questionnaire about their well-being and life habits since the start of the COVID-19 pandemic. This questionnaire was completed during the serology appointment in a separate area ensuring confidentiality, especially from parents. Adolescents were included in this analysis if the three above-mentioned questionnaires were completed (Fig 1).

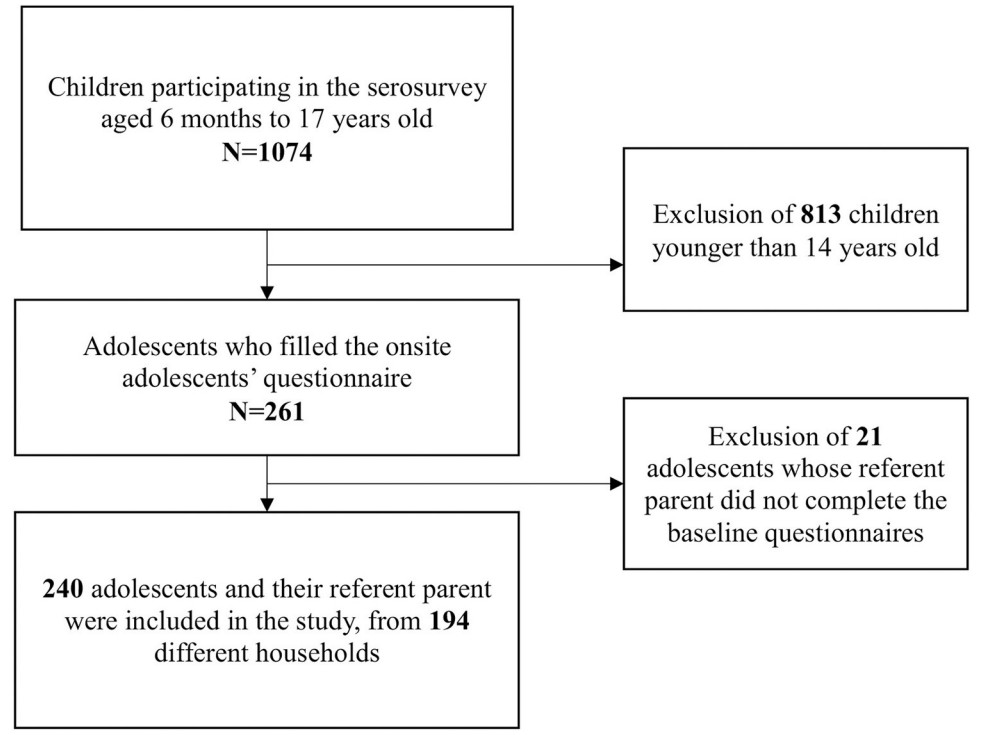

**Fig 1. Study flow diagram.**

## Measures

**HRQoL and psychological distress—adolescent's perception.** Adolescent-reported HRQoL was evaluated over the previous week using the validated French version of the standardized KIDSCREEN-10 scale. The KIDSCREEN-10 evaluates HRQoL with a score from 0 to 100 in children and adolescents aged 8 to 18 years. The internal consistency of the KIDSCREEN-10 index was good ($\alpha$ = 0.81). Adolescents with a HRQoL score lower than one standard deviation below the study population mean were considered as having a low HRQoL [16]. The 2 items of the KIDSCREEN-10 assessing mood and feelings (sadness and loneliness) were used separately as binary proxies for psychological distress. We considered adolescents to feel sad or lonely if they reported feeling so "quite often", "very often" or "always" over the last week.

**HRQoL and psychological distress—parent's perception.** The parent's perception of their children's well-being over the previous week was based on the French version of the KINDL® for parents [17]. This scale measures HRQoL of children and adolescents aged from 7 to 17 years based on their parent's answers, by combining 24 items covering 6 dimensions: physical well-being, emotional well-being, self-esteem, family, friends and school. Adolescents with an overall HRQoL score lower than one standard deviation below the study population mean were considered as having a low HRQoL [16]. Overall internal consistency of the KINDL® was high ($\alpha$ = 0.85). Focusing on the distinct KINDL® dimensions, we defined psychological distress as an emotional well-being score lower than one standard deviation below the study population mean ($\alpha$ = 0.69).

## Outcomes

The 5 outcomes of interest were adolescents' self- and parent-reported HRQoL, as well as self-reported sadness and loneliness, and parent-reported emotional well-being.

**Other variables.** We investigated the following adolescent, parental and household variables: age, sex, total screen time (adolescent-reported number of hours spent on a screen daily), adolescent-reported change in time spent on social media since the start of the pandemic (increase, decrease, no change) and anti-SARS-CoV-2 serological status of the adolescent; age, sex, education, self-perceived mood and anti-SARS-CoV-2 serological status of the referent parent. Values of the anti-SARS-CoV-2 serology ≥0.8 µ/mL were considered positive (Elecsys anti-SARS-CoV-2 S; Roche Diagnostics, Rotkreuz, Switzerland [18]); the serological assessment was conducted before the start of the vaccination campaign in Switzerland. Parent education was measured with a three-level scale: lower (compulsory education), middle (secondary education), and higher (tertiary education). Self-perceived mood of the parent was defined as good if the answer to the question "In general, how would you assess your mood?" was "good" or "very good", and average to poor for answers such as "average", "poor" or "very poor".

Household level variables such as household size, density, financial situation, parents' marital status and family well-being were also included. Household size was expressed as the number of people living in the household. Household density was defined using the measure of crowding from Eurostat; households without a private bedroom for the adolescent(s) were considered crowded [19]. Household financial situation was considered as good if the referent parent answered that they could save money or face minor unexpected expenses, and average to poor if they selected one of the following statements: "I have to be careful with my expenses and an unexpected event could put me into financial difficulty" or "I cannot cover my needs with my income and I need external support". Parents' marital status was dichotomized into married or as couple on one hand and divorced, separated, single or widowed on the other

hand. Family well-being was measured using the family dimension of the KINDL$^{®}$ scale for parents (α = 0.71) [17].

## Statistical analysis

After excluding participants with missing data (N = 15, 6.2%), multivariable models were performed for each outcome as follow: minimal model adjusted for age and sex, or age, sex, financial situation and household density when appropriate, and full model adjusted for adolescents' age, sex, anti-SARS-CoV-2 serological status, total screen time, change in time spent on social media, referent parents' age, sex, anti-SARS-CoV-2 serological status, mood and marital status, as well as household size, density and financial situation. The risk of multi-collinearity was considered acceptable as the variance inflation factor (VIF) was under five. As some of the adolescents were siblings, a generalized estimating equation (GEE) function [20, 21] was used to correct for the familial dependency in the observations with a covariance matrix defined as exchangeable and tests based on sandwich-corrected robust standard errors. Multivariable model results were reported as adjusted odds ratios (aOR) with 95% confidence intervals (95%CI).

Discrepancy between self-reported and parent-reported low HRQoL was coded as a binary variable: 1 if perceptions were different and 0 if they were similar. Risk factors of discrepancy were assessed with GEE adjusting for the above-mentioned covariates, as well as the family well-being score.

Statistical significance was defined at a level of confidence of 95% and all analyses were performed with R (version 4.0.3).

## Results

The sample consisted of 240 adolescents from 194 households. Mean age was 15 years (SD 2.1 years) and 47% were females. Referent parents' mean age was 47 years (SD 9.0 years), 75% being mothers. Table 1 presents a descriptive overview of the adolescents' and their parents' socio-demographic characteristics.

## Adolescent-reported HRQoL, sadness, and loneliness

Overall, 26 (11%) adolescents reported a low HRQoL (Table 1). In the fully adjusted model, referent parents' positive anti-SARS-CoV2 serology (aOR = 3.33; 95%CI: 1.20–9.12) was associated with a lower HRQoL of the adolescent, whereas there was no association with the adolescents' own serological status (P > 0.1; Table 2). Shorter screen time was associated with a lower HRQoL (aOR = 0.89; 95%CI: 0.79–0.98), as was an increase in time spent on social media, although not significant (aOR = 1.60; 95%CI: 0.92–3.70).

Regarding psychological distress, 35% of the adolescents reported feeling sad during the previous week, and 23% felt alone (Table 3). Sadness was more likely among girls, compared to boys (aOR = 3.20; 95%CI: 1.67–6.16), and seemed associated with an increase in the use of social media (aOR = 2.07; 95%CI: 1.08–3.97) and an average to poor household financial situation (aOR = 2.31; IC95%: 1.01–6.10). Sadness was also associated with the referent parent's positive anti-SARS-CoV2 serology (aOR = 2.55; 95%CI: 1.14–5.67), older age (aOR = 1.06; 95%CI: 1.00–1.13) or average to poor mood (aOR = 2.62; 95%CI: 1.10–6.23). Female adolescents (aOR = 4.10; 95%CI: 1.92–8.78) were more likely to report feeling lonely, while a positive anti-SARS-CoV-2 serology was associated with less loneliness (aOR = 0.19; 95%CI: 0.05–0.70) (Table 3). The magnitude of results was similar with the minimally adjusted model (S1 and S2 Tables).

**Table 1. Characteristics of the study population.**

| | Adolescents (N = 240) | Referent parents (N = 194) |
|---|---|---|
| | N (%) | N (%) |
| **Age** in years, mean (SD) (N = 240/N = 194) | 15.2 (2.1) | 46.5 (9.0) |
| **sex** (N = 240/N = 194) | | |
| Male | 128 (53.3) | 48 (24.8) |
| Female | 112 (46.7) | 146 (75.2) |
| **Nationality** (N = 240/N = 194) | | |
| Swiss | 188 (78.3) | 144 (74.2) |
| Portuguese | 10 (4.2) | 8 (4.1) |
| Italian | 14 (5.8) | 14 (7.2) |
| French | 13 (5.4) | 9 (4.6) |
| Others | 15 (6.3) | 19 (9.9) |
| **Ethnicity** (N = 240/N = 194) | | |
| European-Caucasian | 194 (80.1) | 165 (85.1) |
| Other | 46 (19.9) | 29 (14.9) |
| **Parents' marital status** (N = 194) | | |
| Married or in couple | - | 152 (78.4) |
| Divorced, separated, single or widowed | - | 42 (21.6) |
| **Household size** in individuals, mean (SD) (N = 194) | - | 3.2 (1.1) |
| **Education** (N = 194) | | |
| Lower | - | 12 (6.2) |
| Medium | - | 62 (32.0) |
| Higher | - | 120 (61.8) |
| **Household financial situation** (N = 194) | | |
| High | - | 139 (71.6) |
| Average to poor | - | 40 (20.6) |
| Does not want to answer | - | 15 (7.8) |
| **Crowded household** (N = 193) | - | 20 (10.3) |
| **Positive anti-SARS-CoV-2 serology** (N = 240/N = 194) | 53 (22.1) | 45 (23.2) |
| **Low self-reported HRQoL**[1] (N = 235) | 26 (11.0) | - |
| **Self-reported sadness** (N = 236) | 84 (35.5) | - |
| **Self-reported loneliness** (N = 237) | 55 (23.2) | - |
| **Low parent-reported HRQoL**[2] (N = 234) | 28 (11.9) | - |
| **Low parent-reported emotional well-being**[2] (N = 234) | 38 (16.2) | - |
| **Average to poor mood** (N = 194) | - | 30 (15.5) |

Results are N (%), unless stated otherwise. All variables are self-reported, except serology (see Methods). Other ethnicities include Arab, Asian, African, Indian and South-American. HRQoL stands for health-related quality of life.
[1] Based on the KIDSCREEN-10 scale.
[2] Based on the KINDL® scale.

## Parent-reported HRQoL and emotional well-being

Overall, 28 (12%) adolescents presented a low parent-reported HRQoL (Table 1). The latter was only associated with adolescents' sex, girls being more likely to be perceived as having a low HRQoL compared to boys (aOR = 4.31; 95%CI: 1.63–11.10; Table 2). When focusing on psychological distress, 38 (16%) adolescents presented a low emotional well-being according to their parents (Table 1), which was associated with being female (aOR = 2.73; 95%CI: 1.22–6.06), living in a crowded household (aOR = 3.12; 95%CI: 1.10–9.37), and having an average to

**Table 2. Risk factors for adolescents' self- and parent-reported low Health-Related Quality of Life (HRQoL).**

| | | Adolescent-reported HRQoL (KIDSCREEN-10) | | | | | Parent-reported HRQoL (KINDL®) | | | |
|---|---|---|---|---|---|---|---|---|---|---|
| | N | High N (%) | Low N (%) | Low HRQoL aOR (95%CI)[a] | | N | High N (%) | Low N (%) | Low HRQoL aOR (95%CI)[a] | |
| **Age of the adolescent (years)[1]** | 235 | 15.4 (1.6) | 13.8 (4.2) | 0.83 (0.54–0.97) | * | | 15.4 (1.2) | 15.4 (1.1) | 0.95 (0.63–1.41) | |
| **Age of the parent (years)[1]** | 235 | 48.4 (5.1) | 48.5 (7.3) | 1.02 (0.89–1.12) | | | 48.3 (5.5) | 48.7 (4.4) | 1.02 (0.94–1.10) | |
| **Sex of the adolescent** | 235 | | | | | 234 | | | | |
| Boy | | 113 (90.4) | 12 (9.6) | 1 | | | 118 (94.4) | 7 (5.6) | 1 | |
| Girl | | 96 (87.3) | 14 (12.7) | 1.10 (0.59–2.22) | | | 88 (80.7) | 21 (19.3) | 4.31 (1.63–11.10) | ** |
| **Self-perceived mood of the parent** | 235 | | | | | 234 | | | | |
| Good | | 178 (89.9) | 20 (10.1) | 1 | | | 178 (89.4) | 21 (10.6) | 1 | |
| Average to poor | | 31 (83.8) | 6 (16.2) | 1.30 (0.46–5.56) | | | 28 (80.0) | 7 (20.0) | 1.63 (0.51–5.26) | |
| **Financial situation of the household** | 235 | | | | | 234 | | | | |
| Good | | 156 (90.7) | 16 (9.3) | 1 | | | 152 (88.4) | 20 (11.6) | 1 | |
| Average to poor | | 35 (79.5) | 9 (20.5) | 2.30 (0.85–6.23) | | | 36 (83.7) | 7 (16.3) | 1.36 (0.56–5.65) | |
| No answer | | 18 (94.7) | 1 (5.3) | 0.53 (0.07–5.27) | | | 18 (94.7) | 1 (5.3) | 0.67 (0.07–6.67) | |
| **Household size (individuals)[1]** | 235 | 3.2 (1.0) | 3.8 (2.0) | 1.52 (0.79–2.66) | | | 3.3 (1.2) | 3.4 (1.1) | 1.20 (0.79–1.78) | |
| **Household density** | 234 | | | | | 233 | | | | |
| Non-crowded | | 192 (89.7) | 22 (10.3) | 1 | | | 190 (89.2) | 23 (10.8) | 1 | |
| Crowded | | 16 (80.0) | 4 (20.0) | 2.08 (0.48–6.70) | | | 15 (75.0) | 5 (25.0) | 2.05 (0.64–6.61) | |
| **Change in social media habits** | 229 | | | | | 228 | | | | |
| Same or less | | 115 (92.0) | 10 (8.0) | 1 | | | 109 (87.9) | 15 (12.1) | 1 | |
| More | | 88 (84.6) | 16 (15.4) | 1.60 (0.92–3.70) | | | 91 (87.5) | 13 (12.5) | 0.83 (0.62–1.45) | |
| **Screen time (hours)[1]** | 229 | 1.8 (2.0) | 1.6 (1.1) | 0.89 (0.79–0.98) | * | 228 | 1.9 (2.0) | 1.7 (1.3) | 0.95 (0.82–1.06) | |
| **Parent anti-SARS-CoV-2 serology** | 235 | | | | | 234 | | | | |
| Negative | | 163 (92.1) | 14 (7.9) | 1 | | | 159 (90.3) | 17 (9.7) | 1 | |
| Positive | | 46 (79.3) | 12 (20.7) | 3.33 (1.20–9.12) | * | | 47 (81.0) | 11 (19.0) | 2.01 (0.69–6.34) | |
| **Adolescent anti-SARS-CoV-2 serology** | 235 | | | | | 234 | | | | |
| Negative | | 167 (90.8) | 17 (9.2) | 1 | | | 162 (89.0) | 20 (11.0) | 1 | |
| Positive | | 42 (82.4) | 9 (17.6) | 1.08 (0.57–2.27) | | | 44 (84.6) | 8 (15.4) | 1.38 (0.44–4.03) | |
| **Parents' marital status** | 235 | | | | | 234 | | | | |
| Married or in couple | | 163 (88.6) | 21 (11.4) | 1 | | | 160 (87.4) | 23 (12.6) | 1 | |
| Divorced, separated, single or widowed | | 46 (90.2) | 5 (9.8) | 0.84 (0.27–2.20) | | | 46 (90.2) | 5 (9.8) | 0.57 (0.16–2.13) | |

Results are adjusted odds ratios (aOR) and 95% confidence intervals (CI) from multivariable generalized estimating equations adjusted for all covariates in first column.

[a] based on 225 observations.

* indicates $P < 0.05$

** indicates $P < 0.01$.

[1] Descriptive analysis presented as mean (SD); OR applicable for each additional unit of continuous variables.

poor financial situation (aOR = 2.37; 95%CI: 0.98–7.38; Table 3). Results of the minimally adjusted model were of the same magnitude (S1 and S2 Tables).

## Comparison of adolescents' and parents' perceptions

Based on the dichotomous classification of the KIDSCREEN-10 and the KINDL®, adolescents' and parents' perceptions matched in 184 (79%) of cases. However, 22 (9.5%) of the adolescents presented a low HRQoL that did not seem identified by their parents. Misperception was more likely among girls compared to boys (aOR = 2.88; 95%CI: 1.54–5.41) and in households with lower family well-being (aOR = 0.91; 95%CI: 0.86–0.96; Table 4).

**Table 3. Risk factors for adolescents' self-reported loneliness and sadness, and parent-reported low emotional well-being.**

| | | Self-reported sadness | | | | | Self-reported loneliness | | | | | Parent-reported low emotional well-being (KINDL®) | | | |
|---|---|---|---|---|---|---|---|---|---|---|---|---|---|---|---|
| | N | No N(%) | Yes N(%) | Sadness aOR (95% CI)[a] | | N | No N(%) | Yes N(%) | Loneliness aOR (95% CI)[a] | | N | High N(%) | Low N(%) | Low emotional well-being aOR (95% CI)[a] | |
| Age of the adolescent (years)[1] | 236 | 15.4 (1.1) | 14.9 (3.1) | 0.96 (0.72–1.13) | | 237 | 15.3 (1.6) | 14.9 (3.2) | 1.01 (0.80–1.23) | | 234 | 15.4 (1.2) | 15.3 (1.1 | 0.84 (0.62–1.17) | |
| Age of the parent (years)[1] | 236 | 47.8 (4.9) | 49.5 (6.0) | 1.06 (1.00–1.13) | * | 237 | 48.2 (5.2) | 49.3 (6.0) | 1.04 (0.97–1.12) | | 234 | 48.0 (5.2) | 50.2 (5.9) | 1.10 (0.97–1.22) | |
| Sex of the adolescent | 236 | | | | | 237 | | | | | | | | | |
| Boy | | 94 (75.2) | 31 (24.8) | 1 | | | 109 (86.5) | 17 (13.5) | 1 | | 234 | 112 (89.6) | 13 (10.4) | 1 | |
| Girl | | 58 (52.3) | 53 (47.7) | 3.20 (1.67–6.16) | ** | | 73 (65.8) | 38 (34.2) | 4.10 (1.92–8.78) | ** | | 84 (77.1) | 25 (22.9) | 2.73 (1.22–6.06) | * |
| Self-perceived mood of the parent | 236 | | | | | | | | | | | | | | |
| Good | | 138 (69.3) | 61 (30.7) | 1 | | 237 | 158 (79.0) | 42 (21.0) | 1 | | 234 | 168 (84.4) | 31 (15.6) | 1 | |
| Average to poor | | 14 (37.8) | 23 (62.2) | 2.62 (1.10–6.23) | * | | 24 (64.9) | 13 (35.1) | 1.68 (0.71–3.98) | | | 28 (80.0) | 7 (20.0) | 0.94 (0.39–2.30) | |
| Financial situation of the household | 236 | | | | | | | | | | | | | | |
| Good | | 115 (66.9) | 57 (33.1) | 1 | | 237 | 136 (78.6) | 37 (21.4) | 1 | | 234 | 147 (85.5) | 25 (14.5) | 1 | |
| Average to poor | | 22 (48.9) | 23 (51.1) | 2.31 (1.01–6.10) | * | | 30 (66.7) | 15 (33.3) | 1.98 (0.68–4.63) | | | 32 (74.4) | 11 (25.6) | 2.37 (0.98–7.38) | * |
| No answer | | 15 (78.9) | 4 (21.1) | 0.85 (0.30–2.91) | | | 16 (84.2) | 3 (15.8) | 1.41 (0.36–5.50) | | | 17 (89.5) | 2 (10.5) | 0.98 (0.29–4.76) | |
| Household size (individuals)[1] | 236 | 3.3 (1.1) | 3.3 (1.3) | 1.15 (0.97–1.48) | | 237 | 3.3 (1.1) | 3.4 (1.5) | 1.17 (0.67–1.59) | | | 3.4 (1.2) | 3.1 (1.4) | 0.73 (0.39–1.21) | |
| Household density | 235 | | | | | 236 | | | | | 233 | | | | |
| Non-crowded | | 142 (66.0) | 73 (34.0) | 1 | | | 167 (77.3) | 49 (22.7) | 1 | | | 182 (85.4) | 31 (14.6) | 1 | |
| Crowded | | 10 (50.0) | 10 (50.0) | 1.60 (0.58–4.77) | | | 15 (75.0) | 5 (25.0) | 0.99 (0.74–1.65) | | | 13 (65.0) | 7 (35.0) | 3.12 (1.10–9.37) | * |
| Change in social media habits | 230 | | | | | 231 | | | | | 228 | | | | |
| Same or less | | 93 (73.8) | 33 (26.2) | 1 | | | 105 (83.3) | 21 (16.7) | 1 | | | 107 (86.3) | 17 (13.7) | 1 | |
| More | | 55 (52.9) | 49 (47.1) | 2.07 (1.08–3.97) | * | | 72 (68.6) | 33 (31.4) | 1.88 (0.96–3.90) | | | 84 (80.8) | 20 (19.2) | 1.57 (0.67–3.40) | |
| Screen time (hours)[1] | 230 | 1.7 (1.8) | 2.0 (2.0) | 1.04 (0.91–1.27) | | 231 | 1.8 (1.8) | 2.1 (2.1) | 1.06 (0.87–1.27) | | 228 | 1.9 (2.0) | 1.5 (1.5) | 0.98 (0.71–1.15) | |
| Parent anti-SARS-CoV-2 serology | 236 | | | | | 237 | | | | | 234 | | | | |
| Negative | | 121 (68.4) | 56 (31.6) | 1 | | | 139 (78.1) | 39 (21.9) | 1 | | | 148 (84.1) | 28 (15.9) | 1 | |
| Positive | | 31 (52.5) | 28 (47.5) | 2.55 (1.14–5.67) | * | | 43 (72.9) | 16 (27.1) | 2.05 (0.84–5.01) | | | 48 (82.8) | 10 (17.2) | 0.67 (0.23–1.98) | |
| Adolescent anti-SARS-CoV-2 serology | 236 | | | | | 237 | | | | | | | | | |
| Negative | | 116 (63.4) | 67 (36.6) | 1 | | | 135 (73.4) | 49 (26.6) | 1 | | 234 | 153 (84.1) | 29 (15.9) | 1 | |
| Positive | | 36 (67.9) | 17 (32.1) | 0.54 (0.22–1.32) | | | 47 (88.7) | 6 (11.3) | 0.19 (0.05–0.70) | ** | | 43 (82.7) | 9 (17.3) | 1.59 (0.57–4.38) | |

*(Continued)*

**Table 3.** (Continued)

| | N | No N(%) | Yes N(%) | Sadness aOR (95% CI)[a] | N | No N(%) | Yes N(%) | Loneliness aOR (95% CI)[a] | N | High N(%) | Low N(%) | Low emotional well-being aOR (95% CI)[a] |
|---|---|---|---|---|---|---|---|---|---|---|---|---|
| | | **Self-reported sadness** | | | | **Self-reported loneliness** | | | | **Parent-reported low emotional well-being (KINDL[R])** | | |
| **Parents' marital status** | 236 | | | | 237 | | | | 234 | | | |
| Married or in couple | | 120 (65.2) | 64 (34.8) | 1 | | 147 (79.5) | 38 (20.5) | 1 | | 155 (84.7) | 28 (15.3) | 1 |
| Divorced, separated, single or widowed | | 32 (61.5) | 20 (38.5) | 1.17 (0.61–2.20) | | 35 (67.3) | 17 (32.7) | 1.36 (0.58–3.21) | | 41 (80.4) | 10 (19.6)) | 0.84 (0.27–2.20) |

Results are adjusted odds ratio (aOR) and 95% confidence intervals (CI) from multivariable generalized estimating equations adjusted for all covariates in first column.

[a] based on 225 observations

* indicates P < 0.05

** indicates P < 0.01.

[1] Descriptive analysis presented as mean (SD); OR applicable for each additional unit of continuous variables.

# Discussion

In this population-based study conducted during the second COVID-19 wave in Switzerland, 8 to 9 months after the start of the pandemic, we observed that a meaningful proportion of adolescents reported a low HRQoL and some psychological distress, with average HRQoL being lower than pre-pandemic levels [22]. Risk factors for a self-reported low HRQoL included a positive anti-SARS-CoV-2 serology of the referent parent, while being a girl was a risk factor for low HRQoL as perceived by parents. Adolescents' self-reported HRQoL was generally corroborated by their parent's observation. However, girls and adolescents living in households with lower family well-being, were at higher risk of misperception by their parents. When looking at psychological distress, risk factors for adolescent-reported sadness or loneliness included being a girl, an increase in time spent on social media or living in a household with a disadvantaged financial situation, as well as characteristics of the referent parent such as older age, positive anti-SARS-CoV-2 serology and poorer mood.

Compared to the Swiss pre-pandemic reference level where the median of the KIDSCREEN-10 score was 80 [22], adolescents presented a lower HRQoL in this study with a median of 72.5 (MAD = 14.8), which might be partly explained by the impact of the pandemic. This is in line with results from other studies, which reported high levels of adolescent stress, worry and anxiety during the pandemic, likely due to restrictive sanitary measures and the worrying pandemic environment [1]. Estimates of sadness and loneliness among adolescents were lower in the present study compared to other studies [23, 24]. However, most previous studies were based on convenience samples and were not population-based. Furthermore, in Switzerland sanitary restrictions to contain SARS-CoV-2 spread were comparatively less strict than in other OECD countries [13]. Finally, during the study period sanitary restrictions were regularly changing in Geneva, but were never as strict as during the first pandemic wave when schools were closed and all activities suspended [14]. The psychological distress estimates are thus likely to be lower than what would have been observed during the first wave, when most other studies were conducted [23–25].

Lower well-being was associated with being a girl, as generally observed [26, 27]; different factors such as the onset of menstruation, inwards coping patterns and high, sometimes contradictory, social expectations may contribute to this discrepancy among adolescents [28]. Parental lower mood was a risk factor for adolescent psychological distress, which is consistent

**Table 4. Risk factors of discrepancies between adolescents' and parents' perception of adolescents' low HRQoL.**

| | | Adolescent and parent perception of adolescent HRQoL | | | |
|---|---|---|---|---|---|
| | N | Match N (%) | Mismatch N (%) | Perception mismatch aOR (95%CI) | |
| **Age of the adolescent (years)**[1] | 232 | 15.4 (1.2) | 15.2 (1.1) | 0.86 (0.68–1.16) | |
| **Age of the parent (years)**[1] | 232 | 48.4 (5.3) | 48.4 (5.8) | 1.02 (0.96–1.10) | |
| **Sex of the adolescent** | 232 | | | | |
| Boy | | 108 (87.1) | 16 (12.9) | 1 | |
| Girl | | 76 (70.4) | 32 (29.6) | 2.88 (1.54–5.41) | ** |
| **Self-perceived mood of the parent** | 232 | | | | |
| Good | | 161 (81.7) | 36 (18.3) | 1 | |
| Average to poor | | 23 (65.7) | 12 (34.3) | 1.91 (0.67–5.39) | |
| **Financial situation of the household** | 232 | | | | |
| Good | | 135 (78.9) | 36 (21.1) | 1 | |
| Average to poor | | 32 (76.2) | 10 (23.8) | 1.03 (0.68–2.88) | |
| No answer | | 17 (89.5) | 2 (10.5) | 0.37 (0.07–1.90) | |
| **Household size (individuals)**[1] | 232 | 3.2 (1.0) | 3.6 (1.7) | 1.24 (0.85–1.82) | |
| **Household density** | 231 | | | | |
| Non-Crowded | | 172 (81.5) | 39 (18.5) | 1 | |
| Crowded | | 11 (55.0) | 9 (45.0) | 2.51 (0.76–8.09) | |
| **Change in social media habits** | 226 | | | | |
| Same or less | | 99 (80.5) | 24 (19.5) | 1 | |
| More | | 79 (76.7) | 24 (23.3) | 0.88 (0.42–1.88) | |
| **Screen time (hours)**[1] | 226 | 1.9 (2.0) | 1.6 (1.2) | 0.93 (0.87–1.01) | |
| **Serology result of the parent** | 232 | | | | |
| Negative | | 145 (82.9) | 30 (17.1) | | |
| Positive | | 39 (68.4) | 18 (31.6) | 1.72 (0.67–4.37) | |
| **Serology result of the adolescent** | 232 | | | | |
| Negative | | 148 (81.3) | 34 (18.7) | 1 | |
| Positive | | 36 (72.0) | 14 (28.0) | 2.54 (0.98–6.62) | |
| **Family well-being score**[1] | 232 | 66.3 (7.8) | 58.8 (10.0) | 0.91 (0.86–0.96) | ** |

Results are adjusted odds ratio (aOR) and 95% confidence intervals (CI) from multivariable generalized estimating equations adjusted for all covariates in first column based on 225 observations. Perception mismatch is coded as 1 if low HRQoL from adolescents' and parents' perception do not match and 0 otherwise.

* indicates $P < 0.05$

** indicates $P < 0.01$.

[1] Descriptive analysis presented as mean (SD); OR applicable for each additional unit of continuous variables.

with studies showing that parent's mental health directly impacts children's functioning [12]. Living in a crowded household was associated with adolescents' psychological distress, which could be explained by difficulties to maintain privacy and a personal space at home [29]. This aspect has possibly worsened since the start of the pandemic as household members were likely to spend more time at home. On the opposite, household size was not associated with adolescents' well-being, consistent with other findings [30].

In accordance with warnings issued by psychologists, an increase in time spent on social media seemed associated with sadness and loneliness [31]. Causality could be bidirectional. As a consequence of other activities being restricted, adolescents may have felt lonely, sad and bored, and thus have spent more time on social media. Conversely, it may have increased their exposure to alarming and contradictory information and affected their well-being [32]. Interestingly, overall screen time was not related to psychological distress and positively associated

with HRQoL. It suggests that examining the type of online activities may be of more significance than the overall screen time [33].

Another unexpected finding was that adolescents' low HRQoL was associated with a positive anti-SARS-CoV-2 serology of the parent but not with their own serological result. This may reflect that adolescents were more impacted by their relatives health than by their own during the COVID-19 pandemic [34]. It may also mirror the impact of difficult circumstances linked with a parent being infected by SARS-CoV-2 [35]. More broadly, it is in line with an extensive body of literature showing the negative impact of parental illness on children psychological well-being [36]. Adolescents' negative anti-SARS-CoV-2 serology was associated with loneliness. A possible explanation could be that adolescents respecting social distancing measures more carefully were less infected but also felt lonelier, although this remains speculative at this stage.

An association between poor socioeconomic conditions and adolescents' psychological distress was probably already present before the COVID-19 pandemic [37]. However, its significance was likely to increase as sanitary measures may have resulted in fewer quiet and safe spaces and more financial instability for adolescents, particularly among vulnerable ones [38].

Self- and parent-reported HRQoL matched in most cases. However, 9.5% of the adolescents presented a low HRQoL that did not seem to be identified by their parents. Misperception was more likely among girls and adolescents living in families with lower family well-being; these adolescents may represent a vulnerable group whose mental health issues are under-recognized. This finding is meaningful as parents' awareness is important for the early detection of psychological distress among their children and for care seeking [39].

Early intervention is needed to improve adolescents' well-being, especially for girls and those living in disadvantaged households. Indeed, these adolescents seem particularly at risk for high psychological distress, which is also more likely to remain unidentified by their parents. Measures could include free psychological consultations for adolescents [40] or online social support [5, 41]. The use of such resources by our study population was not explored and is worth further investigations. In view of our results, it also seems important that the increase in time spent on social networks during the pandemic does not become established as a new habit. Measures could promote and facilitate access to alternative leisure such as sport, art and music.

This study presents several strengths. First, it relies on a randomly selected population-based sample. Thanks to the family-based design, data comes from both adolescents and their parents, which enables comparisons of both perceptions. Finally, previous SARS-CoV-2 infection is assessed with an objective measure. The study also presents several limitations. First, it is a cross-sectional study relying on self-reported data. It does not allow us to firmly conclude whether the reported adolescents' low HRQoL is caused by the pandemic. The sample size is rather small, which limits statistical power. The participation rate was quite low and despite the random selection process, disadvantaged socioeconomic groups were underrepresented in our sample, as commonly observed in such studies [42]. Therefore estimates of low well-being and psychological distress may have been underestimated. Finally, we did not study potential protective factors of adolescents' well-being and psychological distress, such as physical activity or family cohesion. Further studies should focus on these aspects to inform the design of effective prevention and mitigation measures.

## Conclusion

Amid the COVID-19 second wave in Geneva, a meaningful share of adolescents reported low well-being; their overall HRQoL level showed a decrease compared with pre-pandemic

estimates. Adolescents living in unfavourable family environments, including crowded households, poor financial situation or distressed parents, seemed particularly affected. Tailored measures, especially targeting these vulnerable adolescents, are needed given the uncertainties related to the progression of the pandemic. Finally, monitoring is necessary to evaluate the long-term effects of the pandemic on adolescents' mental health. Indeed, some of the impacts may be temporary thanks to the lifting of restrictions, while others may persist over time, especially among vulnerable individuals.

## Supporting information

**S1 Table. Risk factors for adolescents' self- and parent-reported low Health-Related Quality of Life (HRQoL), minimally and fully adjusted models.** N = 225.
(PDF)

**S2 Table. Risk factors for adolescents' self-reported loneliness and sadness, and parent-reported low emotional well-being, minimally and fully adjusted models.** (N = 225).
(PDF)

## Acknowledgments

We are grateful to the staff of the Unit of Population Epidemiology of the HUG Division of Primary Care Medicine as well as to all the participants whose contributions were invaluable to the study.

We also acknowledge the contribution of all the members of the Specchio-COVID19 study group: Isabelle Arm-Vernez, Andrew S Azman, Fatim Ba, Oumar Ba, Delphine Bachmann, Jean-François Balavoine, Michael Balavoine, Hélène Baysson, Lison Beigbeder, Julie Berthelot, Patrick Bleich, Gaëlle Bryand, François Chappuis, Prune Collombet, Delphine Courvoisier, Alain Cudet, Carlos de Mestral Vargas, Paola D'ippolito, Richard Dubos, Roxane Dumont, Isabella Eckerle, Nacira El Merjani, Antoine Flahault, Natalie Francioli, Marion Frangville, Idris Guessous, Séverine Harnal, Samia Hurst, Laurent Kaiser, Omar Kherad, Julien Lamour, Pierre Lescuyer, François L'Huissier, Fanny-Blanche Lombard, Andrea Jutta Loizeau, Elsa Lorthe, Chantal Martinez, Lucie Ménard, Lakshmi Menon, Ludovic Metral-Boffod, Benjamin Meyer, Alexandre Moulin, Mayssam Nehme, Natacha Noël, Francesco Pennacchio, Javier Perez-Saez, Giovanni Piumatti, Didier Pittet, Jane Portier, Klara M Posfay-Barbe, Géraldine Poulain, Caroline Pugin, Nick Pullen, Zo Francia Randrianandrasana, Aude Richard, Viviane Richard, Frederic Rinaldi, Jessica Rizzo, Khadija Samir, Claire Semaani, Silvia Stringhini, Stéphanie Testini, Guillemette Violot, Nicolas Vuilleumier, Ania Wisniak, Sabine Yerly, María-Eugenia Zaballa.

## Author Contributions

**Conceptualization:** Roxane Dumont, Viviane Richard, Hélène Baysson, Elsa Lorthe, Stephanie Schrempft, Rémy P. Barbe, Klara M. Posfay-Barbe, Idris Guessous, Silvia Stringhini.

**Data curation:** Roxane Dumont, Viviane Richard.

**Formal analysis:** Roxane Dumont, Viviane Richard.

**Funding acquisition:** Hélène Baysson, Ania Wisniak, Idris Guessous, Silvia Stringhini.

**Investigation:** Hélène Baysson, Giovanni Piumatti, Klara M. Posfay-Barbe, Idris Guessous, Silvia Stringhini.

**Methodology:** Roxane Dumont, Viviane Richard, Elsa Lorthe, Ania Wisniak.

**Project administration:** Idris Guessous, Silvia Stringhini.

**Resources:** Hélène Baysson, Klara M. Posfay-Barbe, Silvia Stringhini.

**Supervision:** Hélène Baysson, Elsa Lorthe, Giovanni Piumatti, Stephanie Schrempft, Rémy P. Barbe, Idris Guessous.

**Validation:** Elsa Lorthe, Giovanni Piumatti, Rémy P. Barbe, Silvia Stringhini.

**Visualization:** Viviane Richard, Giovanni Piumatti, Stephanie Schrempft, Ania Wisniak.

**Writing – original draft:** Roxane Dumont, Viviane Richard, Elsa Lorthe.

**Writing – review & editing:** Roxane Dumont, Viviane Richard, Hélène Baysson, Elsa Lorthe, Giovanni Piumatti, Stephanie Schrempft, Ania Wisniak, Rémy P. Barbe, Klara M. Posfay-Barbe, Idris Guessous, Silvia Stringhini.

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
