## [Decision Letter · Decision Letter 0]

21 Jun 2022

PONE-D-22-08850Determinants of adolescents’ Health-Related Quality of Life and psychological distress during the COVID-19 pandemicPLOS ONE

Dear Dr. Stringhini,

Thank you for submitting your manuscript to PLOS ONE. After careful consideration, we feel that it has merit but does not fully meet PLOS ONE’s publication criteria as it currently stands. Therefore, we invite you to submit a revised version of the manuscript that addresses the points raised during the review process.

We look forward to receiving your revised manuscript.

Kind regards,

Cecilia Acuti Martellucci, M.D.

Academic Editor

PLOS ONE

Journal Requirements:

Additional Editor Comments:

Please address the concerns expressed by the reviewers.

Reviewers' comments:

Reviewer's Responses to Questions

**Comments to the Author**

1. Is the manuscript technically sound, and do the data support the conclusions?

Reviewer #1: Partly

Reviewer #2: Yes

2. Has the statistical analysis been performed appropriately and rigorously? 

Reviewer #1: No

Reviewer #2: Yes

3. Have the authors made all data underlying the findings in their manuscript fully available?

Reviewer #1: Yes

Reviewer #2: Yes

4. Is the manuscript presented in an intelligible fashion and written in standard English?

Reviewer #1: Yes

Reviewer #2: Yes

5. Review Comments to the Author

Reviewer #1: This study examined the psychological well-being of adolescents in relation to COVID-19 infection and socioeconomic status. The topic is intriguing. The sample size is not large but acceptable because it reached statistical significance. I have some comments below.

Major

・Although the authors adopted many variables, my suspicion is multicollinearity. I would control for confounding more carefully. For example, the authors showed the association between parents’ Covid-19 positive and a lower HRQoL of the adolescent. I would control for the economic status in this association because it is associated with both HRQoL and risk for infection.

On another note, I would conduct principal component analyses to see how many components exist in these variables.

・Regarding the question about sadness and loneliness, my concern is that they are self-assessments only. I wonder if some have a psychiatric diagnosis (such as developmental disorder or attachment disorder). The authors are encouraged to control for the diagnosis.

・Since a child's well-being is also affected by the family environment, it would be better to consider not only income but also, for example, whether the child is a single parent, whether the child is adequately educated, and the level of nurturing.

・The authors found being a girl was a risk factor for the sadness. If so, I would classify the children as not both sexes together but each sex separately because boys and girls would have a different distribution of well-being. Identifying being a girl as a risk factor is, in my opinion, an oversimplification of their aim is to assess the determinants of adolescents’ Health-Related Quality of Life and psychological distress (self-reported and parent-reported) during COVID-19.

Reviewer #2: Dear authors,

Thank you for giving me the opportunity to review this interesting manuscript on the impact of the COVID-19 pandemic on adolescents wellbeing.

While the introduction and methods are clear and well written, from my point of view the discussion has to be better developed: authors do not detail enough the practical implications of their findings.

It would be nice to know their insights on how to further develop clinical guidelines or future projects on these issues.

Best regards and thanks

6. PLOS authors have the option to publish the peer review history of their article (what does this mean?). If published, this will include your full peer review and any attached files.

Reviewer #1: **Yes: **Yuta Y Aoki

Reviewer #2: No

---

## [Author Response · Author response to Decision Letter 0]

12 Jul 2022

Dear Editor,

On behalf of my co-authors, I am pleased to submit our revised manuscript entitled “Determinants of adolescent’s Health-Related Quality of Life and psychological distress during the COVID-19 pandemic” (Manuscript PONE-D-22-08850R1).

We would like to thank the editor and the reviewers for their careful reading and judicious comments that helped us improve this work. We have now addressed all of the reviewers’ concerns and journal requirements and feel the manuscript has much improved as a consequence. 

Point-by-point responses (see the attached file)

Page and line numbers refer to the version of the manuscript with revisions.

Response: The manuscript was adapted according to the PLOS ONE guidelines.

Response: Due to the fact that we are reporting information on a relatively small sample of adolescents from a specific geographic location, our data contains potentially identifying information, despite all precautions taken. Therefore, the study steering committee members decided to make these data accessible to researchers who meet the criteria for access to confidential data upon reasonable request for data sharing to the Unit of Population Epidemiology (uep@hcuge.ch). All requests will be evaluated by the Data Access Committee and approved on the basis of their scientific quality.

Response: The ethics statement was put in the Methods section.

Comments from Reviewer #1

This study examined the psychological well-being of adolescents in relation to COVID-19 infection and socioeconomic status. The topic is intriguing. The sample size is not large but acceptable because it reached statistical significance. I have some comments below.

1. Major. Although the authors adopted many variables, my suspicion is multicollinearity.

Response: We thank the Reviewer for this important comment. In addition to a careful descriptive analysis of the covariates that was conducted before running the models, we computed the Variance inflation Factor, a widely used statistical tool that identifies multicollinearity. The Variance Inflation Factor of each covariate was considered acceptable as values were under 5, confirming that there was no multicollinearity (Table 1).

Thus, although our covariates may represent close concepts, multicollinearity did not seem to be an issue in our models. Accordingly, the following statement was included:

“The risk of multicollinearity was considered acceptable as the variance inflation factor (VIF) was under five.” (page 9, line 241-242) 

2. I would control for confounding more carefully. For example, the authors showed the association between parents’ Covid-19 positive and a lower HRQoL of the adolescent. I would control for the economic status in this association because it is associated with both HRQoL and risk for infection.

Response: We thank the review for this constructive remark. The association between the anti-SARS-CoV-2 serological status and the HRQoL of the adolescent was adjusted for socio-economic conditions. However, we agree with the Reviewer’s suggestion about careful confounding control. Accordingly, we ran minimal models only adjusted for age and sex, or age, sex, household financial situation and crowding index when appropriate. Estimates were similar compared with the fully adjusted model and added in the Appendix. We edited the Methods and Results sections as follow:

“[…multivariable models were performed for each outcome] as follow: minimal model adjusted for age and sex, or age, sex, financial situation and household density when appropriate, and full model adjusted for […].” (page 9, lines 236-237)

“In the fully adjusted model, […]” (page 11, lines 271-272)

“The magnitude of results was similar with the minimally adjusted model (Appendix 1 and 2).” (page 12, line 288)

“Results of the minimally adjusted model were of the same magnitude (Appendix 1 and 2).” (page 12, lines 297-298)

3. On another note, I would conduct principal component analyses to see how many components exist in these variables.

Response: We thank the Reviewer for the interesting suggestion. Indeed, the included covariates may have close underlying concepts. When performing the Factor Analysis for Mixed Data (as covariates are mixed, continuous and discrete), it appears that parent’s and adolescent’s serology, as well as the financial situation vary together on the first dimension (Figure 1). On the other hand, it is mostly explained by the adolescent’s age and time spent on social media. However, the variability explained by the factorial analysis remains relatively small, and as confirmed by the VIF above, the correlation between the covariates is not high enough for multicollinearity to be an issue.

Fig 1. Factor Analysis for Mixed Data of the models’ covariates

4. Regarding the question about sadness and loneliness, my concern is that they are self-assessments only. I wonder if some have a psychiatric diagnosis (such as developmental disorder or attachment disorder). The authors are encouraged to control for the diagnosis.

Response: Thank you for this comment. Our goal was to evaluate adolescents’ general well-being and psychological distress, and to better understand how adolescents feel during the pandemic. Thus, we believe that self-report is adequate in this regard. Furthermore, the point of view of another informant was provided with the parent-reported adolescent’s emotional well-being. As acknowledged by the reviewer, controlling for existing psychiatric diagnosis would have been relevant. However, in our sample, only 3 adolescents were reported to have a psychiatric disorder; making it impossible to control for this. 

5. Since a child's well-being is also affected by the family environment, it would be better to consider not only income but also, for example, whether the child is a single parent, whether the child is adequately educated, and the level of nurturing.

Response: We fully agree with the Reviewer on this important point. In addition to the financial situation, parent mood was also included in our models to take the family environment into account. Following the Reviewer’s comment, we further added the parents’ marital status as a covariate. There was no significant association with any of our five outcomes. Although very interesting, child-rearing and nurturing were not assessed in our study and could not be included in the analyses. The Methods section was edited with the additional variable, as were the tables 2 and 3 in the Results section.

“Parents’ marital status was dichotomized into married or as couple on one hand and divorced, separated, single or widowed on the other hand.” (page 9, lines 230-231)

“[…] full model adjusted for […] referent parents’ […] marital status […]. (page 9, lines 237-240)

6. The authors found being a girl was a risk factor for the sadness. If so, I would classify the children as not both sexes together but each sex separately because boys and girls would have a different distribution of well-being. Identifying being a girl as a risk factor is, in my opinion, an oversimplification of their aim is to assess the determinants of adolescents’ Health-Related Quality of Life and psychological distress (self-reported and parent-reported) during COVID 19.

Response: Thank you for making this point. Stratifying our analyses by sex could have given an additional insight into the association between sex and health-related quality of life, but it was not possible due to the small sample size. Also, we believe that a study specifically designed to answer this question would be more appropriate to understand this relationship. However, we agree that we could discuss further the fact that girl’s health-related quality of life is lower compared to boys. The discussion was developed accordingly.

“[…] different factors such as the onset of menstruation, inwards coping patterns and high, sometimes contradictory, social expectations may contribute to this discrepancy among adolescents.” (page 17, lines 47-49)

Comments from Reviewer #2

Dear authors, 

Thank you for giving me the opportunity to review this interesting manuscript on the impact of the COVID-19 pandemic on adolescents’ wellbeing. While the introduction and methods are clear and well written, from my point of view the discussion has to be better developed: authors do not detail enough the practical implications of their findings. It would be nice to know their insights on how to further develop clinical guidelines or future projects on these issues. 

Best regards and thanks

Response: We thank the Reviewer for making this important point. We elaborated on the potential measures that could be implemented to protect adolescents’ well-being. We also acknowledge that further studies are needed to adequately identify relevant protective factors and interventions. The discussion section has been edited accordingly.

“[Early intervention is needed to improve adolescents’ well-being,] especially for girls and those living in disadvantaged households. Indeed, these adolescents seem particularly at risk for high psychological distress, which is also more likely to remain unidentified by their parents. […] In view of our results, it also seems important that the increase in time spent on social networks during the pandemic does not become established as a new habit. Measures could promote and facilitate access to alternative leisure such as sport, art and music.“ (pages 18-19, lines 88-95)

“Finally, we did not study potential protective factors of adolescents’ well-being and psychological distress, such as physical activity or family cohesion. Further studies should focus on these aspects to design effective prevention and mitigation measures.” (page 19, lines 106-109)

---

## [Decision Letter · Decision Letter 1]

29 Jul 2022

Determinants of adolescents’ Health-Related Quality of Life and psychological distress during the COVID-19 pandemic

PONE-D-22-08850R1

Dear Dr. Stringhini,

We’re pleased to inform you that your manuscript has been judged scientifically suitable for publication and will be formally accepted for publication once it meets all outstanding technical requirements.

Kind regards,

Cecilia Acuti Martellucci, M.D.

Academic Editor

PLOS ONE

Additional Editor Comments (optional):

Reviewers' comments:

Reviewer's Responses to Questions

**Comments to the Author**

1. If the authors have adequately addressed your comments raised in a previous round of review and you feel that this manuscript is now acceptable for publication, you may indicate that here to bypass the “Comments to the Author” section, enter your conflict of interest statement in the “Confidential to Editor” section, and submit your "Accept" recommendation.

Reviewer #1: All comments have been addressed

2. Is the manuscript technically sound, and do the data support the conclusions?

Reviewer #1: Yes

3. Has the statistical analysis been performed appropriately and rigorously? 

Reviewer #1: Yes

4. Have the authors made all data underlying the findings in their manuscript fully available?

Reviewer #1: No

5. Is the manuscript presented in an intelligible fashion and written in standard English?

Reviewer #1: Yes

6. Review Comments to the Author

Reviewer #1: (No Response)

7. PLOS authors have the option to publish the peer review history of their article (what does this mean?). If published, this will include your full peer review and any attached files.

Reviewer #1: No

---

## [Editor Report · Acceptance letter]

3 Aug 2022

PONE-D-22-08850R1 

Determinants of adolescents’ Health-Related Quality of Life and psychological distress during the COVID-19 pandemic 

Dear Dr. Stringhini:

I'm pleased to inform you that your manuscript has been deemed suitable for publication in PLOS ONE. Congratulations! Your manuscript is now with our production department. 

Kind regards, 

on behalf of

Dr. Cecilia Acuti Martellucci 

Academic Editor

PLOS ONE